# GoBigger: A Scalable Platform for Cooperative-Competitive Multi-Agent Reinforcement Learning

**Ming Zhang[1]\*, Shenghan Zhang[1]\*, Zhenjie Yang[1]\*, Lekai Chen[1]\*,**
**Jinliang Zheng[1], Chao Yang[2], Chuming Li[1], Hang Zhou[1], Yazhe Niu[2], Yu Liu[1,2]†**
[1]SenseTime Research, [2]Shanghai AI Laboratory, Shanghai, China
`{zhangming,zhangshenghan,yangzhenjie,chenlekai,zhengjinliang,`
`zhouhang2}@sensetime.com`
`{lichuming,niuyazhe,liuyu}@pjlab.org.cn`

## Abstract

The emergence of various multi-agent environments has motivated powerful algorithms to explore agents' cooperation or competition. Even though this has greatly promoted the development of multi-agent reinforcement learning (MARL), it is still not enough to support further exploration on the behavior of swarm intelligence between multiple teams, and cooperation between multiple agents due to their limited scalability. To alleviate this, we introduce GoBigger, a scalable platform for cooperative-competition multi-agent interactive simulation. GoBigger is an enhanced environment for the Agar-like game, enabling the simulation of multiple scales of agent intra-team cooperation and inter-team competition. Compared with existing multi-agent simulation environments, our platform supports multi-team games with more than two teams simultaneously, which dramatically expands the diversity of agent cooperation and competition, and can more effectively simulate the swarm intelligent agent behavior. Besides, in GoBigger, the cooperation between the agents in a team can lead to much higher performance. We offer a diverse set of challenging scenarios, built-in bots, and visualization tools for best practices in benchmarking. We evaluate several state-of-the-art algorithms on GoBigger and demonstrate the potential of the environment. We believe this platform can inspire various emerging research directions in MARL, swarm intelligence, and large-scale agent interactive learning. Both GoBigger and its related benchmark are open-sourced. More information could be found at https://github.com/opendilab/GoBigger.

## 1 Introduction

The swarm behavior of multi-agent systems (MAS) widely exists in nature and human society. In MAS, individual agent pursues their goal and interacts with each other in local areas, following the rules of cooperation or competition, and then the intelligent behavior of the agent group forms the complex collective behaviors. The phenomena of collective behaviors can be found in the flocking birds (Bhattacharya & Vicsek, 2010), molecular motors (Chowdhury, 2006), human crowds (Helbing et al., 2000), and traffic systems (Kanagaraj & Treiber, 2018). To understand and simulate such phenomena, some rule-based models (Castellano et al., 2009) can simulate the swarm behavior in an unconstrained environment with random movement. However, in a complex interactive environment such as intra-cellular molecular motor transport, where the interaction of agents is time-varying and updatable, it is challenging to recover the underlying collective behaviors by manually designing the controllers or rules.

Interactive simulation of multi-agent systems can provide significant convenience for multi-agent learning algorithms. Some existing multi-agent simulation environments mainly focus on the coop-

---

*Denotes equal contributions.
†Corresponding author

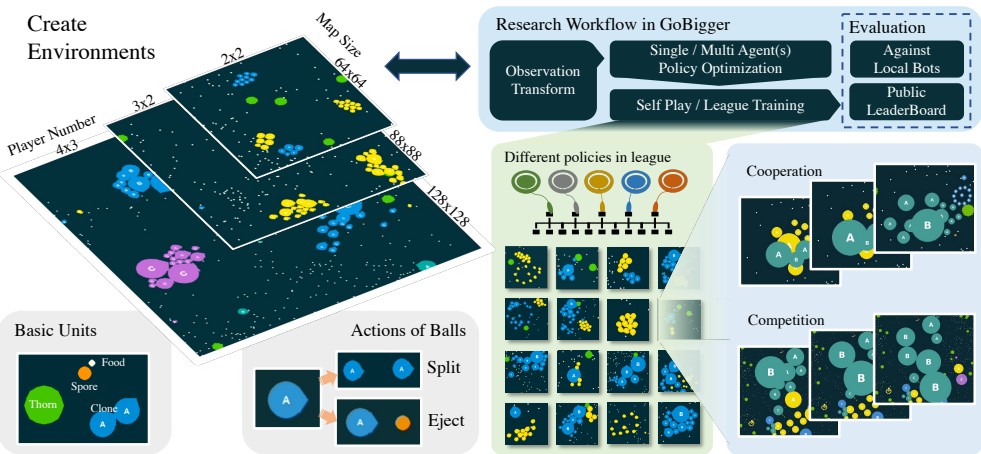

Figure 1: New users can follow the given research workflow, while advanced users can customize the configuration of the environment and define new tasks based on GoBigger. The lower left part shows the basic units (balls) and the related actions. The lower right part shows the training in the league with many games going on at the same time, from where cooperative and competitive behaviors of agents can be observed.

eration (Xuan et al., 2001) between agents, such as Bi-DexHands (Chen et al., 2022), Multi-Agent MuJoCo (Peng et al., 2021), PressurePlate McInroe (2022), RWARE (Papoudakis et al., 2021), and Flatland (Mohanty et al., 2020). These environments offer simulation designs for cooperation between agents but do not care about multi-agent competition. With the rapid development of the field of multi-agent reinforcement learning, more and more multi-agent simulation environments focus on both cooperation and competition, including LBF (Papoudakis et al., 2020), MALMO (Johnson et al., 2016), DM Lab (Beattie et al., 2016), DM Lab2D (Beattie et al., 2020), Derk's Gym Norén (2020), PommerMan (Resnick et al., 2018), StarCraft Multi-Agent Challenge (SMAC) (Samvelyan et al., 2019), Google Research Football (Kurach et al., 2019), Multi-Agent Particle Environment (MPE) (Mordatch & Abbeel, 2017), Hide and Seek (Baker et al., 2019), and Neural MMO (Suarez et al., 2021). Agents in these environments are divided into different teams to achieve intra-team cooperation and inter-team competition. However, most of them consist of up to two teams, named $2 \times N$ mode. That means they can not handle situations where multiple teams cooperate or compete with each other, which is necessary for the research on the swarm behavior of multi-agent systems. Besides, the performance gap caused by different levels of cooperation is not significant in most environments including SMAC, Google Research Football, and NeuralMMO. For the most popular multi-agent environment SMAC, recently SMACv2 (Ellis, 2022) shows that agents can get high performance after dropping teammates' observations. The author uses only the identity and observation of agents in the training phase under QMIX (Rashid et al., 2018) and MAPPO (Yu et al., 2021) to achieve the same performance as that using the global state additionally. And the replay videos of these environments with well-trained agents could not show the complex cooperation. For those environments that emphasize cooperation through game mechanics, the observations of them are too simple that different multi-agent algorithms can easily reach high performance. More details about the various multi-agent simulation environments are shown in Table 1.

In this paper, we propose a scalable platform, GoBigger, aiming to delve into cooperative-competitive multi-agent reinforcement learning for swarm intelligence between multiple teams. Different from the previous multi-agent simulation environment, GoBigger is a scalable environment that enables the simulation of various teams and agents in each team. In other words, in the $M \times N$ game mode of GoBigger, $M$ means the number of teams in the environment, and $N$ means the number of players in each team. This new game mode dramatically expands the way of agent cooperation and competition and can more effectively simulate the swarm intelligent agent behavior. In addition, GoBigger makes agents with intra-team cooperation and inter-team competition achieve higher performance according to the restriction of game mechanics and rules, which is approved in Section 6.1. We offer a diverse set of challenge scenarios in GoBigger for best practices in benchmarking. In most of the given scenarios, each player in a team is controlled by an independent agent that has to act based on only its local observation or all teammates' observations. Meanwhile, GoBigger is a more complex

Table 1: Comparison of GoBigger with other related multi-agent simulation environments. *Agents Size* denotes agent scale in the interactive environment, where $M$ and $N$ mean the number of competitive teams and the agent number in each team, respectively, in the $M \times N$ settings. *Actions* denotes whether the action space is continuous, discrete, or hybrid. *Obs* denotes whether to support the partial observation. $+$ in *Coop* and *Comp* indicates the importance of cooperation and competition. — means there is no cooperation or competition in the environment.

|  | Agents Size | Actions | Obs | Coop | Comp |
|---|---|---|---|---|---|
| SMAC (Samvelyan et al., 2019) | $2 \times N$ | Discrete | Partial | + | +++ |
| GRF (Kurach et al., 2019) | $2 \times N$ | Discrete | Partial | + | +++ |
| Bi-DexHands (Chen et al., 2022) | $1 \times N$ | Discrete | Full | +++ | — |
| MPE (Mordatch & Abbeel, 2017) | $2 \times N$ | Hybrid | Full | ++ | ++ |
| Hide-and-Seek (Baker et al., 2019) | $2 \times N$ | Discrete | Partial | ++ | ++ |
| MAMujoco (Peng et al., 2021) | $1 \times N$ | Continuous | Partial | +++ | — |
| NeuralMMO (Suarez et al., 2021) | $M \times N$ | Discrete | Partial | + | +++ |
| **GoBigger (Ours)** | $M \times N$ | Hybrid | Partial | +++ | +++ |

environment with well-designed action space and observation space, giving partial view for all agents in the game. A reproducible benchmark including several state-of-the-art algorithms under different scenarios is accessible for users to quickly get started with GoBigger and demonstrate the potential of the environment. GoBigger also features game systems configurable for users to research on individual aspects of intelligence and on combinations thereof, with given rule-based built-in bots and visualization tools to make it easier for users to evaluate their agents. The overview of the research workflow for users is shown in Figure 1.

## 2    RELATED WORK

Most multi-agent reinforcement learning environments are mainly designed for cooperation or competition between no more than two teams of agents. SMAC (Samvelyan et al., 2019) consists of a set of carefully designed micro scenarios and necessitates learning one or more micromanagement techniques to defeat the enemy. But the SOTA algorithms can achieve high scores even without cooperation between agents. Google Research Football (Kurach et al., 2019) is a novel open-source reinforcement learning environment provides different modes for both single-agent and multi-agent settings, while there is no baseline indicating multi-agent cooperation and competition in the environment. Furthermore, both SMAC and Google Research Football can not handle situations where multiple (more than two) teams in a game, making them unable to be used for the research on swarm behaviors. Agar (staghuntrpg) implemented with python is also available and is evaluated on recent works (Tang et al., 2021). However, it focuses on cooperation and competition between two players but not multiple teams, and its scenarios are not complex. Neural MMO (Suarez et al., 2021) is a massively multi-agent environment for AI research, which support spans 1 to 1024 agents and minute- to hours-long time horizons. It also allows $M \times N$ game mode in its scenarios, and agents need to cooperate with teammates to fight against other teams in the map. According to its given baselines, we find that agents in Neural MMO focus more on competition with other agents but less on cooperation up to now. And open-sourced agents in the challenge are more likely to play on their own instead of cooperating with teammates.

## 3    DESIGN PRINCIPLES AND SYSTEM OVERVIEW

GoBigger is inspired by the popular online multiplayer game Agar (SA). In a regular full game of Agar with OFA(One For All) mode, players cooperate with their teammates and fight against other teams for the final champion. On this basis, many modifications have been made to allow more

cooperation and competition in GoBigger. Each player, which is represented by one or more balls (dubbed **clone ball**), increases its size by colliding and merging with other balls within a bounded rectangular area in a limited time. The larger the size of clone balls, the higher the player's score. At the beginning of the game, a single controllable clone ball (one player) is randomly spawned on the map. In addition to the clone balls, there are three uncontrollable neutral balls in the game map (dubbed as **food ball**, **spore ball**, and **thorn ball**), as shown in Figure 1.

## 3.1 BASIC UNITS

**Food Ball:** Food balls are neutral resources and they are always fixed on the map. Clone balls can eat food balls, and then absorb their size. Food balls will be randomly and continuously generated until the number of available food balls reaches the maximum.

**Spore Ball:** Spore balls are ejected by clone balls. They will stay on the map and can be eaten by any other clone balls.

**Thorn Ball:** Thorn balls are another kind of neutral resource in the game. Different from food balls, they have a larger size and less quantity on the map. In addition, they can eat the spore balls ejected from clone balls and absorb their size. When a clone ball eats a thorn ball, it will evenly split into several small balls. It means a player can push a thorn ball to an opponent's clone ball, force it to split, and eat its small balls.

**Clone Ball:** A player consists of several clone balls and they are the only controllable balls. In addition, a clone ball can eat other smaller balls including food balls, thorn balls, spore balls, and clone balls (from teammates or opponents) by covering their center. Clone balls of an agent can be merged after a certain time. Clone balls will decay in size to ensure they will not grow too large. If all clone balls of a player are eaten, this player will respawn on the map randomly in the next frame. A player can control clone balls by the following three actions in total:

- **Move:** Moving can help a clone ball eat other balls or escape from danger. The bigger the clone ball, the slower it moves.

- **Eject:** Ejecting a spore ball can help a clone ball drop some size and move faster (Figure 1). When a clone ball ejects, a new spore ball will appear on the clone ball's moving direction at high speed but quickly slow down and stop.

- **Split:** Splitting helps a clone ball to evenly split into two clone balls (Figure 1). Besides, a player can have at most 16 clone balls. To move faster, a player can split and turn the clone balls into smaller ones and get a higher speed.

## 3.2 GAME SYSTEMS

The base game representation is a limited map comprising the units declared in Section 3.1. Note that the map is not a grid map, which means that the continuous state is a complex challenge for agents. At the same time, GoBigger makes the operating efficiency of the game acceptable for users. The whole game can be split into the following systems: resources, cooperation, and competition.

**Resources:** Resource system is designed for navigation and multi-task reasoning. In GoBigger, food balls and thorn balls can help the agents to grow larger. Eating food balls is safe but slow while eating thorn balls is dangerous but quick. There are food balls and thorn balls randomly distributed on the map. The agent needs to decide which ball to eat according to its current size and the state of other agents around it to obtain faster development. The initial state and the regeneration rates of resources are all configurable.

**Cooperation:** Cooperation system is designed for direct cooperation among agents in a team. For a single agent, the maximum number of clone balls is 16 (configurable), which causes an agent cannot move fast by splitting indefinitely on its own. At the same time, each clone ball of the agent has a 20 seconds cooldown period, which is reset after each split. When the clone ball is on its cooldown period, it cannot merge with other clone balls that belong to the same agent. This means that splitting is dangerous because failing to merge quickly can easily lead to attacks by other agents. The above two constraints make multiple agents in the same team need to cooperate and transfer size to each other by splitting and ejecting to move faster and ensure safety. In addition, the final ranking of

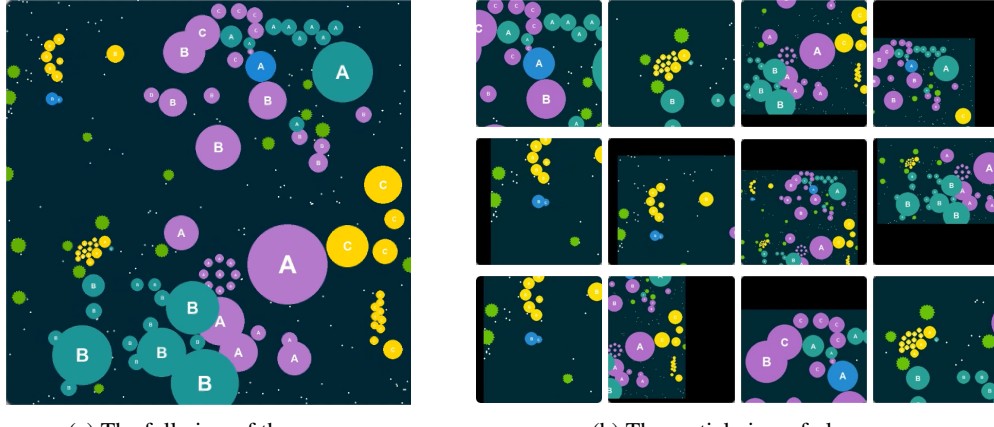

(a) The full view of the game        (b) The partial view of players

Figure 2: Screenshots for the different observation methods of GoBigger. (a) The full view of the game with 4 teams and 3 players in each team. (b) The players in (a) have their observations, and the three figures in each column represent the three players' view from each team.

GoBigger is based on the size of the whole team. Therefore, frequent and reasonable cooperation within the team is an important condition for victory.

**Competition:** Competition system is designed for direct competition among different teams. In GoBigger, the agent loses some size after each timestep. This is more obvious when the agent is very large. Therefore, as the resources on the map decrease as the game progress, the agent needs to eat other agents to ensure it size. Agents can attack other agents by moving and splitting to gain their size. In addition, gathering the size of the team to fight against other agents can often have an advantage in the competition. GoBigger encourages competition among teams.

## 4 ENVIRONMENT

GoBigger provides a set of standard game environment configurations. To facilitate access to the current observation and the efficient control of the balls, GoBigger follows the design of *gym* (Brockman et al., 2016) and develops easy-to-use interfaces of observation and action.

### 4.1 OBSERVATION

As mentioned in Section 3, the observation space of GoBigger is quite complex. It consists of *GlobalState* and *PlayerState*:

*GlobalState* is shared by all players. It includes all essential global information, such as map size, total frames of the game, current running frame in the game, and the leaderboard which consists of the current scores and rankings of all teams.

*PlayerState* is player-wise. It is the information that can be observed within the view of each player, making GoBigger a partially observable MDP (POMDP) (Drake, 1962; Sondik, 1971). Specifically, the player's view depends on the range of the positions of its clone balls. Figure 2 shows the view of different players in a game. When the clone balls of a player are dispersed enough, a larger view will be applied to this player. Given a view, *PlayerState* contains the positions and radius of all balls in the view, including food balls, thorn balls, spore balls, and clone balls, as well as their owner.

More details can be found in Appendix A.1.

### 4.2 ACTION SPACE

The action space of the clone ball consists of move, eject and split could be shown as:

$$a := (x, y, t). \tag{1}$$

Table 2: GoBigger challenges. *Agents Size* denotes the number of agents on the map as Table 1. *Map Size* denotes the length and width of the map. *Food* denotes the range of food balls. *Thorn* denotes the range of thorn balls. *Init Size* denotes the size of clone balls at their first birth. *Limited Frame* denotes the number of frames in a game.

| Name | Agents Size | Map Size | Food | Thorn | Init Size | Limited Frame |
|---|---|---|---|---|---|---|
| **Small maps** | | | | | | |
| st_t1p1 | $1 \times 1$ | $32 \times 32$ | [65, 75] | [1, 2] | 1000 | 3600 (3min) |
| st_t1p2 | $1 \times 2$ | $48 \times 48$ | [130, 150] | [2, 3] | 1000 | 3600 (3min) |
| st_t2p1 | $2 \times 1$ | $48 \times 48$ | [130, 150] | [2, 3] | 1000 | 3600 (3min) |
| st_t2p2 | $2 \times 2$ | $64 \times 64$ | [260, 300] | [3, 4] | 13000 | 3600 (3min) |
| st_t3p2 | $3 \times 2$ | $88 \times 88$ | [500, 560] | [5, 6] | 13000 | 3600 (3min) |
| **Large maps** | | | | | | |
| st_t4p3 | $4 \times 3$ | $128 \times 128$ | [800, 900] | [9, 12] | 1000 | 14400 (12min) |
| st_t5p3 | $5 \times 3$ | $128 \times 128$ | [900, 1000] | [10, 12] | 1000 | 14400 (12min) |
| st_t5p4 | $5 \times 4$ | $144 \times 144$ | [900, 1000] | [10, 12] | 1000 | 14400 (12min) |
| st_t6p4 | $6 \times 4$ | $144 \times 144$ | [1000, 1100] | [11, 13] | 1000 | 14400 (12min) |

Here $a$ denotes the action, $(x, y)$ is a unit vector for the direction of movement, and $t$ is the action type for the clone ball to be executed in the current frame. When a player applies a direction $(x, y)$, enforcement in this direction will be applied on all this player's clone balls and smoothly alters their movement direction to this direction in several frames. The action types consist of moving, ejecting, and splitting. Agents can complete development and attack according to the combination of different action types. For more details, please refer to Appendix A.2.

### 4.3 REWARD

In Gobigger, the goal of all teams is to become bigger and bigger and finally become the biggest. Thus a natural reward for a single agent is to consider its score change of two adjacent steps:

$$Reward_t := Score_{t+1} - Score_t. \tag{2}$$

The above reward can directly reflects the players' growth and induce agents to grow bigger. But it can't encourage agents to cooperate and confront each other very well. Therefore, GoBigger provides some other kinds of reward functions as follows:

**Example 1:** Rapid development must come from eating a large number of thorn balls, so the reward function can be designed as the number of thorn balls eaten by the agent between two adjacent steps.

**Example 2:** The cooperation among teammates consists of ejecting spore balls to each other and eating teammates' clone balls, so the reward function can be designed as the number of spores and the number of clone balls eaten by teammates between two adjacent steps.

**Example 3:** To improve the offensive performance of the agent, the reward function can be designed as the number of clone balls that the agent eats from other teams between two adjacent steps.

With the help of different kinds of rewards and constraints, an agent with strong cooperation and aggression will be obtained. For more details of reward functions, please refer to Section 5.

### 4.4 SCENARIOS

GoBigger consists of a set of scenarios that aim to evaluate the learning cooperation and competition of different agents in different game settings. The scenarios are carefully designed with different settings including agents size, map size, food number, thorn number, and so on. Most of the scenarios are a confrontation among over two teams of players. In different scenarios, agents need to learn to acquire resources on the map and fight against other teams through cooperation. Teams are only different in their locations of birth. When the specified time limit is reached, the team with the

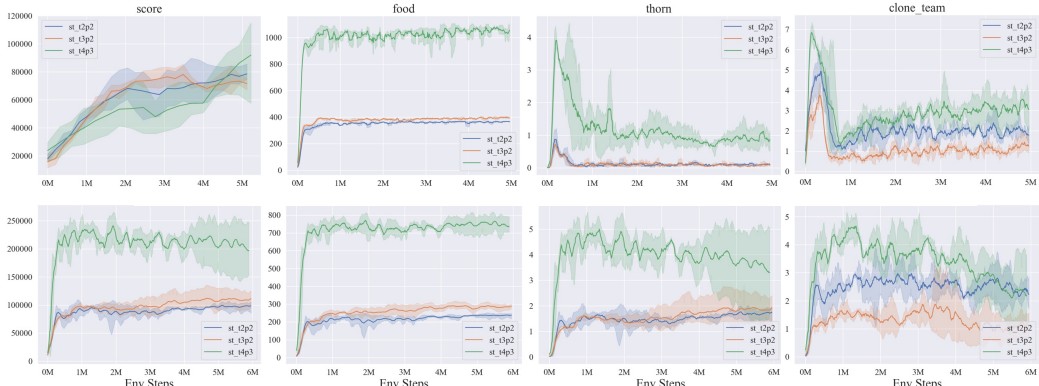

Figure 3: Agents trained on *st_t2p2*, *st_t3p2*, and *st_t4p3* with DQN (the first row) and PPO (the second row). Each column (left to right) denotes the score against built-in bots, the average number of eaten food balls, the average number of eaten thorn balls, and the average number of eaten clone balls from teammates.

higher score wins. In the evaluation phase, agents have to battle against other teams of agents in the scenarios. Built-in bots with different levels could be also chosen as competitors. The rule-based bots can increase their sizes quickly by eating the food balls and thorn balls and avoid being eaten by other players through the given rules. The complete list of challenges is presented in Table 2. The *Init Size* of *st_t2p2* and *st_t3p2* is 13000 to reduce farming time and make the agent more focused on cooperation and competition. According to the map size and the number of agents on the map, the scenarios are simply divided into two parts including small maps and large maps. The large maps may be more difficult as its larger observation while playing against more teams of agents.

## 5 EXPERIMENTS

In this section, we will present some experiments results of GoBigger, which are based on the most representative scenarios: *st_t2p2*, *st_t3p2* and *st_t4p3*. Details of observation encoding and neural network can be found in Appendix B.1. The agents learned by different RL algorithms are evaluated by fighting against the built-in bots of level 3 A.5 several times. The average score will be one of the most important metrics as it stands for the game level of the agents, showing cooperation and competition skills.

### 5.1 SINGLE AGENT ALGORITHMS

We apply DQN (Mnih et al., 2015) and PPO (Schulman et al., 2017) on GoBigger, where each agent only use its own observation and learns a value function independently. Training details are presented in Appendix B.3. The experiment results in different scenarios can be found in Figure 3. The score represents the game level of the agents. Agents can get high scores by eating more food balls and thorn balls in a limited time. It also shows that the popular single-agent algorithms can quickly converge in the given scenarios, inferring that GoBigger is a feasible and challenging environment for multi-agent reinforcement learning research.

### 5.2 MULTI AGENT ALGORITHMS

We also apply several state-of-the-art multi-agent algorithms including QMIX (Rashid et al., 2018), MAPPO (Yu et al., 2021), COMA (Foerster et al., 2018), and VMIX (Su et al., 2021) on GoBigger. The training details are presented in Appendix B.3. Figure 4 shows the performance of different multi-agent algorithms in GoBigger. Overall MAPPO achieves the highest scores when fighting against the built-in bots and is the best performer in all three scenarios with the fewest environment steps. Additionally, with a fixed number of steps, COMA has the worst performance of all algorithms, demonstrating the on-policy policy gradient methods can not have an advantage over different algorithms. Figure 4 also shows that the number of food balls and thorn balls grows rapidly in the training phase. This is in line with the expectations of the environments, in which the fastest way for an agent to farm is to eat food balls and thorn balls quickly. Besides, after enough steps, the agents

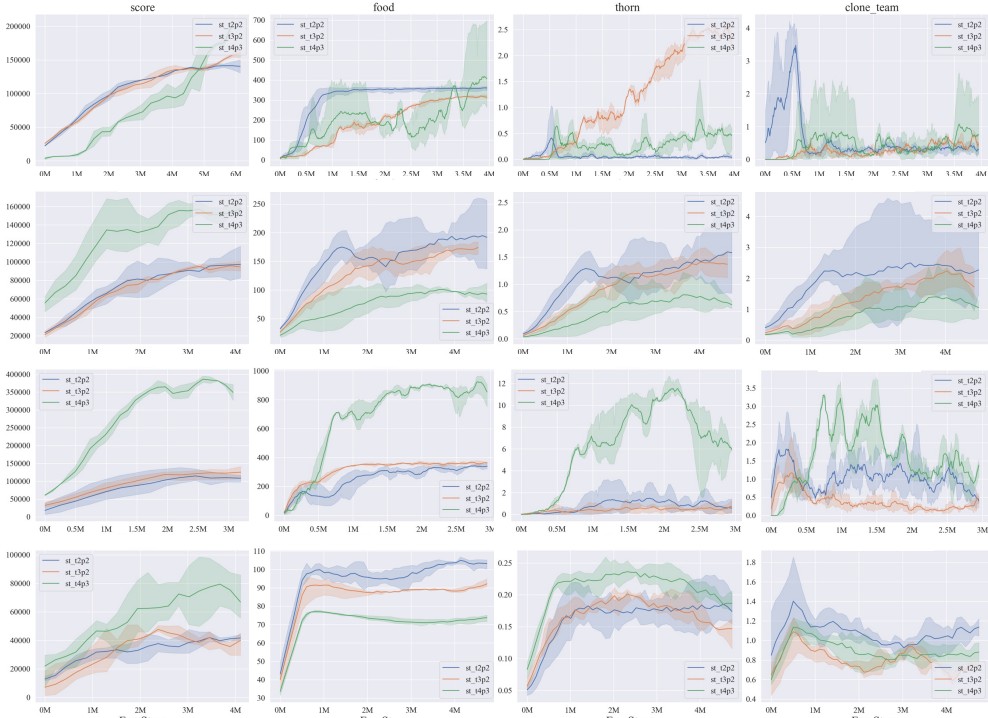

Figure 4: Agents trained on *st_t2p2*, *st_t3p2*, and *st_t4p3* with QMIX (the first row), VMIX (the second row), MAPPO(the third row), and COMA (the fourth row). Each column (left to right) denotes the score against built-in bots, the average number of eaten food balls, eaten thorn balls, and eaten clone balls from teammates.

learn to cooperate with teammates, which is shown from the number of *clone_team*. The growth of *clone_team* means that the agents are trying to merge with teammates.

The results in different scenarios are also shown in Figure 4. *st_t2p2* is a small map with only two teams and four agents, while *st_t4p3* is a large map containing four teams and each consists of three agents. QMIX plays well on small maps including *st_t2p2* and *st_t3p2* but plays bad on large map *st_t4p3*. We infer that the map of *st_t4p3* is too large and too difficult for agents to find their teammates, which causes a low level of cooperation.

## 6    ABLATION STUDY

GoBigger provides small and large maps for training on different scales. Here we explore some factors that have impacts on the performance of an agent, including the necessity of the observations from teammates, the frequency of acting in the game, and the ladder system.

### 6.1    DROPPING OBSERVATIONS FROM TEAMMATES

To explore the importance of cooperation in GoBigger, we drop the observations from teammates, making the agents decide their actions only based on their observations. Figure 5 shows that the agents with teammates' observations outperform the agents without observations from the whole team. Agents can not easily find their teammates without the observations sharing especially on a large map, i.e., *st_t4p3*, resulting that they could not cooperate with their teammates to fight against other teams, which is consistent with our original intention of designing the GoBigger. Appendix C.1 shows more details.

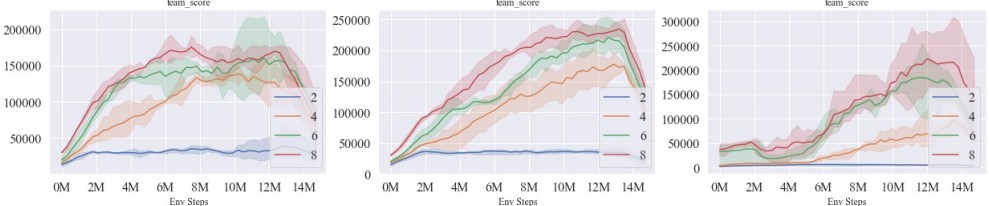

Figure 5: Comparing agents w/ or w/o observation from teammates. Each picture (left to right) denotes the performance of agents on *st_t2p2*, *st_t3p2*, and *st_t4p3*.

Figure 6: Comparing agents with different frequencies of actions on *st_t2p2*, *st_t3p2*, and *st_t4p3* (left to right). Each line denotes the score against bots with a certain frequency of actions.

## 6.2 FREQUENCY OF ACTIONS

In most reinforcement learning environments, agents will execute actions immediately after environments step(Jia et al., 2021). However, it is not necessary in a real-time game. For example, AlphaStar(Vinyals et al., 2019) uses a delay head in their network to predict delay of the next action. To save computing resources, we explore the impact of the frequency of actions on an agent's performance. Figure 6 shows that the agents with $frequency = 8$ outperform other settings. We infer this is because higher frequency will bring more redundant information, which is a little more difficult for agents to understand and distinguish.

## 6.3 LADDER SYSTEM AND LEADERBOARD

In multiplayer games, reward versus some specific opponents is not enough to fully measure agents skills in different situations. So we introduce a ladder system based on *TrueSkill* (Herbrich et al., 2006) to evaluate different kinds of agents objectively. This ladder system will make agents continuously play against each other, and then use their final ranks in each game to update their trueskill scores. Figure 11 shows the ladder system including all of our trained agents as well as built-in bot. More details of the ladder system could be found in Appendix C.2.

## 7 CONCLUSION AND FUTURE WORK

In this paper, GoBigger is presented as a scalable platform for multi-agent interactive simulation. GoBigger allows $M \times N$ game mode that focuses on intra-team cooperation and inter-team competition. We offer a diverse set of challenge scenarios in GoBigger for best practices in benchmarking. A reproducible benchmark including several state-of-the-art algorithms under different scenarios is accessible. GoBigger also features a game system configurable with given rule-based built-in bots and visualization tools to make it easier for users to evaluate their agents.

In the near future, we aim to explore more based on GoBigger. Considering the scenarios in GoBigger now are monotonous, we plan to develop more interesting and complex scenarios that require a higher level of coordination amongst agents. We also plan to expand GoBigger as a massive environment to host thousands of agents. With harder multi-agent coordination and competition problems, we aim to motivate further research in this domain, particularly in areas such as multi-agent exploration and coordination.

**Acknowledgement.** This work was supported by the National Key RD Program of China under Grant STI 2030—Major Projects 2021ZD0201300.

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

# A  GOBIGGER

## A.1  OBSERVATION

The observation in GoBigger is divided into two parts, including **Global State** and **Player States**. After each step, GoBigger will return observation for all agents in the environment. Here are the details for **Global State**. *border* denotes the map size of the game. $total\_frame$ denotes the number of total frames in the game. $last\_frame_{c}ount$ denotes the number of frames that have passed. *leaderboard* denotes the ranks and scores of all agents.

```
1 global_state = {
2    'border': [map_width, map_height],
3    'total_frame': total_frame,
4    'last_frame_count': last_frame_count,
5    'leaderboard': { team_name: team_size }
6 }
```

The **Player States** declare the specific states for each player (or agent) in each team. $player\_id$ denotes the identity of the player in the game. *rectangle* denotes the position of partial vision. *overlap* denotes all kinds of balls in the vision of the player, including the position, radius, score, velocity, direction, player_id, and team_id. $team\_name$ denotes the team's name of the player. *score* denotes the score of the player. $can\_eject$ denotes if the player can eject at this frame. $can\_split$ denotes if the player can split at this frame.

```
1 player_states = {
2    player_id: {
3       'rectangle': [left_top_x, left_top_y, right_bottom_x,
            right_bottom_y],
4       'overlap': {
5          'food': [[position.x, position.y, radius, score], ...],
6          'thorns': [[position.x, position.y, radius, score, vel.x,
                vel.y], ...],
7          'spore': [[position.x, position.y, radius, score, vel.x, vel.
                y, owner], ...],
8          'clone': [[[position.x, position.y, radius, score, vel.x,
                vel.y, direction.x, direction.y, player_id, team_id],
                ...],
9       },
10      'team_name': team_name,
11      'score': player_score,
12      'can_eject': bool,
13      'can_split': bool,
14   },
15   ...
16 }
```

The original structural observations may be difficult for new users in GoBigger. That's why we modified the observations into a simpler tensor array in Section B.1. New users can directly use the given observation encoder, while advanced users can build their own observation encoders which are more suitable for their algorithms.

## A.2  ACTION

In Section 4.2, we offers a simple action space for the control of agents. And the format of the given action could be shown as:

$$a := (x, y, t). \tag{3}$$

The value of $t$ is in $0, 1, 2$, separately representing move, eject and split. Especially, when $t$ is 1 or 2, the clone balls will eject or split in the given direction decided by $(x, y)$. At most time, a player has more than one clone ball. Under this circumstance, the action will be applied to all the clone balls of

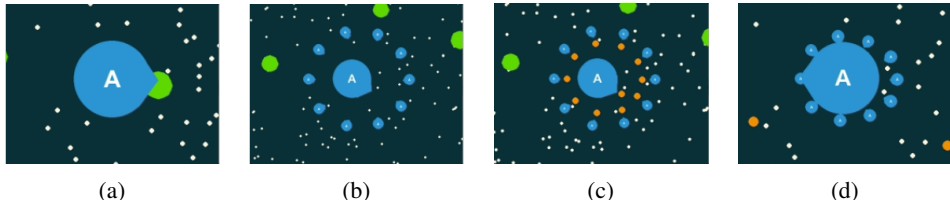

|        |        |        |        |
|:------:|:------:|:------:|:------:|
| (a)    | (b)    | (c)    | (d)    |

Figure 7: Screenshots for one of the action combinations named middle-ejecting. This action combination will make agents gather their size into one big clone ball. (a) The clone ball is ready to eat the thorn ball. (b) After eating the thorn ball, the clone ball is forced to split into several pieces. (c) The agent ejects to the center. (d) The clone balls at the center eat the spore and merges their size together.

the player, i.e., all clone balls with sizes exceeding the split threshold will split when $t$ is 1, and all clone balls with sizes exceeding the eject threshold will eject spores when $t$ is 2.

### A.3 ACTION COMBINATIONS

Though the action space in GoBigger is simple, the combinations of different actions can lead to quick growth, attack, and escape. Figure 7 shows one of the action combinations named middle-ejecting, which is the combination of moving and ejecting. It is really useful to gather most of the size in one clone ball after eating the thorn ball. Besides, agents could use this action combination when they are split into too many pieces, which may help them to avoid being eaten by other agents. In the experiments in Section 5 we find that the agents can perform these action combinations.

### A.4 COLLISION DETECTION

The core problem of GoBigger is how to detect the collision of the balls in each frame and help the game update the state of the balls. The problem of collision detection could be simplified into a scenario where we need to get the collided balls from the given gallery balls for each ball in the given query balls. GoBigger has designed four collision detection algorithms as follows:

**Exhaustive** For each ball in the query, we enumerate each ball in the gallery to determine whether there is a collision. This is an exhaustive solution that will take a long time if the number of balls in the gallery is large. The complexity of the algorithm could be written in:

$$O(n * m) \tag{4}$$

where $m$ denotes the number of balls in the query, and $n$ denotes the number of balls in the gallery.

**Precision** Precision approximation algorithm divides the map into several rows according to the accuracy that has been set, dynamically maintain the row information in each frame and search by row. First, we need to sort the balls in each row according to their ordinate. For the balls in query, we abstract the boundary of the ball into a rectangle, then traverse each row in the rectangle, and find the first ball covered by the query through dichotomy in each row, and then Enumerate the balls in sequence until the ordinate exceeds the boundary of the query rectangle. The complexity of the algorithm could be written in:

$$O(n * \log(n) + \Sigma r * \log(n) + p) \tag{5}$$

where $m$ denotes the number of balls in query, $n$ denotes the number of balls in the gallery, $k$ denotes the precision we set, $r$ denotes the number of balls whose position status has changed and $p$ denotes the number of balls that collide with other balls.

**RebuildQuadTree** We build a *Quadtree* on a two-dimensional plane in every frame based on the positions of balls and search the collisions that happen among the balls in query and gallery according to the *Quadtree*. With the *Quadtree*, GoBigger can quickly find the nearby balls of the given query

ball, which helps to search the collisions in a relatively small search area. The complexity of the algorithm could be written in:

$$O(n * \log(n) + m * \log(n) + p) \tag{6}$$

where $m$ denotes the number of balls in query, $n$ denotes the number of balls in the gallery, and $p$ denotes the number of balls that collide with other balls.

**RemoveQuadTree**    This algorithm is based on the *RebuildQuadTree* A.4, and we add the operations of deleting nodes in the tree with dynamically maintaining the tree. The complexity of the algorithm could be written in:

$$O(r * \log(n) + m * \log(n) + p) \tag{7}$$

where $m$ denotes the number of balls in query, $n$ denotes the number of balls in the gallery, $r$ denotes the number of balls whose position status has changed and $p$ denotes the number of balls that collide with other balls.

To test the efficiency of the above algorithms, we modify the parameters including the number of balls in query and gallery, the number of changed balls, and the iteration rounds to get a more fair and intuitive result. The data in Table 3 comes from the most representative scenarios. We finally choose *Precision* A.4 as our default algorithm for collision detection.

Table 3: Comparison of the different algorithms of collision detection on different settings. *T* denotes the number of balls in the gallery. *Q* denotes the number of balls in the query. *C* denotes the number of changing balls, which means the number of collisions.

|  | T=3000 Q=300 C=600 | T=3000 Q=300 C=1500 | T=10000 Q=1000 C=2000 | T=10000 Q=2000 C=5000 | T=30000 Q=600 C=3000 |
|---|---|---|---|---|---|
| Exhaustive | 688ms | 1067ms | 8384ms | 12426ms | 127000ms |
| Precision | 14ms | 16ms | 61ms | 86ms | 403ms |
| Rebuild QuadTree | 47ms | 50ms | 339ms | 586ms | 5691ms |
| Remove QuadTree | 48ms | 178ms | 497ms | 2460ms | 8419ms |

### A.5    BUILT-IN BOTS

Built-in bots of different levels can help users to get started with the environment and perform a standard evaluation for algorithm development. **Level 1** The bot of level 1 aims to collect neutral resources on the map for quick development in size. It can only move and eat the closest food balls in its view. **Level 2** We add the exploration of thorn balls in view for the bot of level 2. The bot prefers to eat thorn ball in its view. **Level 3** The bot of level 3 can avoid being eaten by other larger players as they will move far away from the larger player in their view.

## B    EXPERIMENTS

### B.1    OBSERVATION TRANSFORM

It is challenging to encode thousands of entities compared to other multi-agent environments. In order to better model the observation information, we transform the original information into **Scalar info**, **Spatial info**, **Team info**, **Ball info**. As shown in (Figure 8), Scalar Info models the size of the agent's local vision, the current ranking and score, and the action type at the last moment. Spatial info models the location of different types of balls in vision. Team info models information of teammates in vision. Ball info models the properties of different balls, including relative position, size, and speed. Here are more details of the information after transformation.

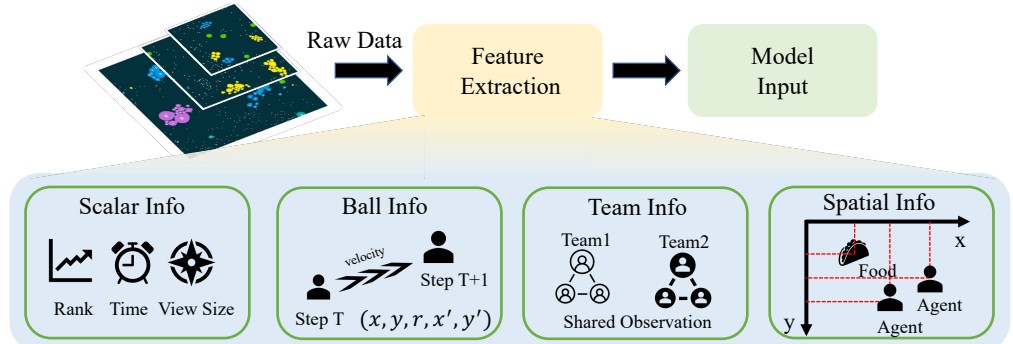

Figure 8: The overview of feature encoder.

**Scalar info**

- (view_x, view_y): The positions of the center of the observation.
- view_width: The width of the observation.
- score: Score of the agent.
- team_score: Team score of the agent.
- time: Remaining game time.
- rank: Rank of the agent.
- last_action_type: Action type in the previous step.

**Ball info**

- alliance: The numbering for different types in observation.
- score: Scores for different types of balls.
- radius: Radius for different types of balls.
- rank: Rank for different types of balls.
- x,y: position for different types of balls.
- next_x,next_y: The predicted position in the next frame for different types of balls.
- ball_num: The number of the balls(clone and thorn) in observation.

**Team info**

- alliance: The teammate's identity information.
- view_x,view_y: The teammate's position information.
- player_num: The number of the teammates.

**Spatial info**

- food_x, food_y: The position of food balls.
- spore_x, spore_y: The position of spore balls.
- ball_x, ball_y: The position of clone balls and thorn balls.
- food_num: The number of food balls.
- spore_num: The number of spore balls.

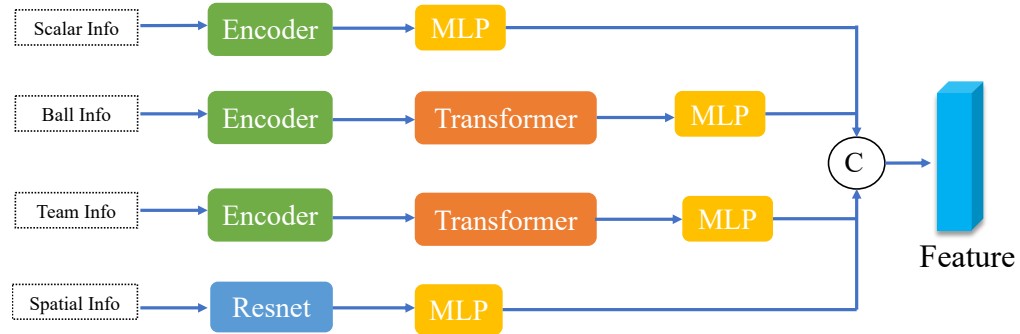

Figure 9: Neural network in experiments.

## B.2 Neural Network

The neural network in our experiments consists of MLP (Haykin, 1994), Transformer (Vaswani et al., 2017), and ResNet (He et al., 2016). The specific structure is shown in Figure 9. Finally, we concatenate all the output embeddings into a feature.

## B.3 Experimental Setup

**DQN** The *batch_size* is set as 512. *learning_rate* is 1e-4, *replay_buffer_size* is 2e4.

**PPO** The *batch_size* is set as 256. *clip_range* is 0.2, *gae_lambda* is 0.95, and *grad_clip* is 0.5.

**QMIX** The *batch_size* is set as 512. *learning_rate* is 1e-4, *replay_buffer_size* is 2e4.

**MAPPO** The *batch_size* is set as 512. *clip_range* is 0.2, *gae_lambda* is 0.95, and *grad_clip* is 0.5.

**VMIX** The *batch_size* is set as 2048, and the data in every step contains all the observations and actions of all the agents. *td_lambda* is 0.95, and *grad_clip* is 0.5.

**COMA** The *batch_size* is set as 2048, and the data in every step contains all the observations and actions of all the agents. *td_lambda* is 0.95, *target_update_interval* is 20, and *grad_clip* is 0.5.

## C Ablation Study

### C.1 Dropping Observations from Teammates

Figure 10 shows the *food*, *thorn*, *spore*, *clone_self*, *clone_team*, and *clone_other* of the experiments in Section 6.1. With the observations from teammates, agents are more likely to eat thorn balls instead of food balls. We can find that the number of eaten thorn balls is more than that without the observations, as eating thorn balls can provide more scores. A single agent can not eat too many thorn balls, as it will split into several pieces after eating a thorn ball. With the cooldown period, an agent can not merge quickly to gather enough size to eat the next thorn ball. But when an agent learns to cooperate with teammates, they can gather size quickly and eat the next thorn ball ignoring the cooldown period, which could be found by the number of *clone_team* in the figure. Eating more clone balls from teammates means that the agents can better cooperate with other agents in the same team.

### C.2 Ladder System and Leaderboard

Our ladder system is based on *TrueSkill* (Herbrich et al., 2006), which is a skill-based ranking system developed by Microsoft for use with video game matchmaking on Xbox Live. TrueSkill is designed to support games with more than two players. In our ladder system, we assume that each contestant

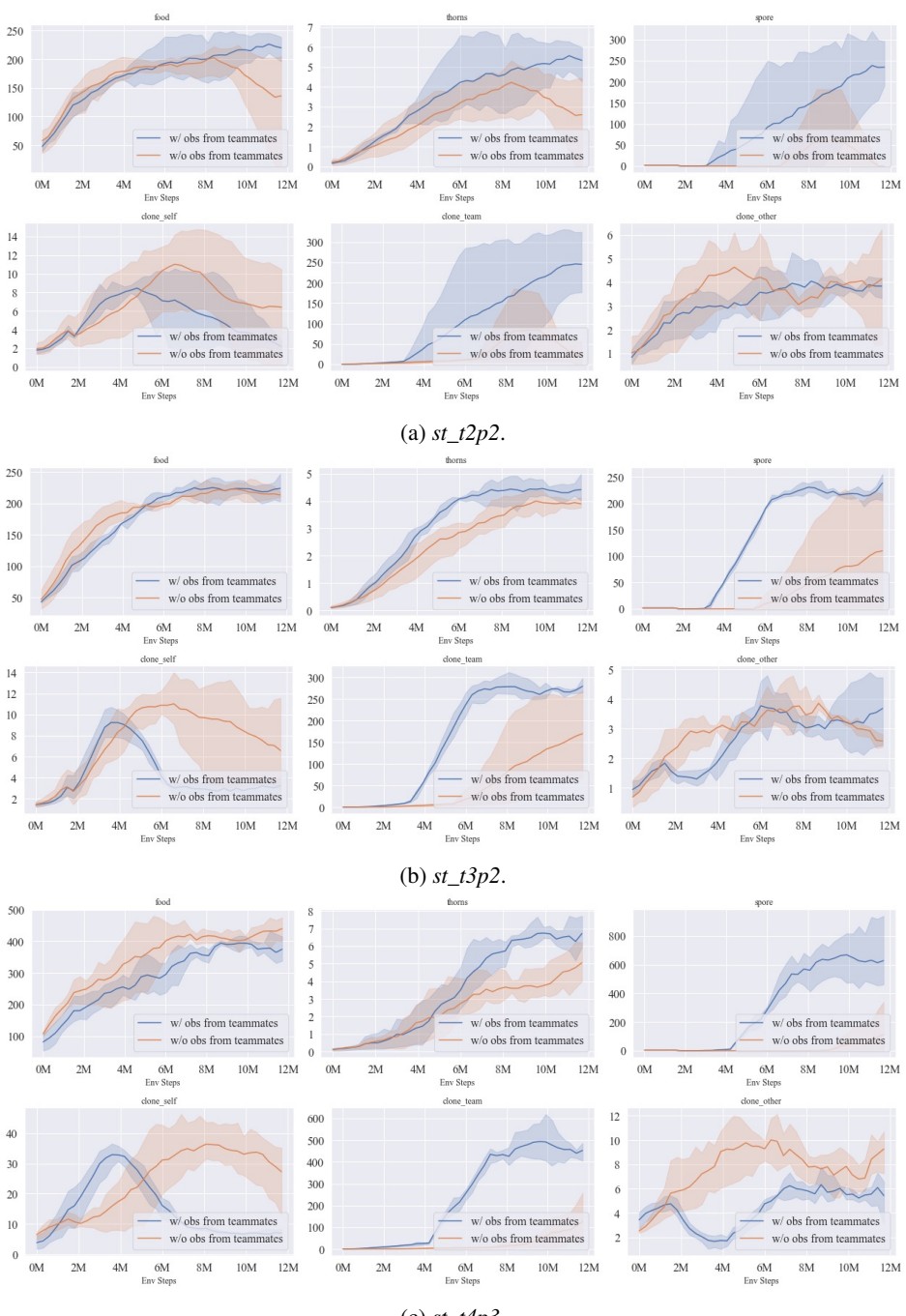

(a) *st_t2p2*.

(b) *st_t3p2*.

(c) *st_t4p3*.

Figure 10: Comparing agents w/ or w/o observation from teammates.

controls a whole team. The contestant continuously plays against other contestant and update their scores based on the ranking at the end of the game. The initial score for each contestant is 1000. To ensure the balance of contestants' matches, we select the contestant with the least number of matches each time. Eq 8 shows the quality we use as the indicator for selecting opponents.

$$quality_{draw}(\beta^2, \mu_i, \mu_j, \sigma_i, \sigma_j) = \sqrt{\frac{2\beta^2}{2\beta^2 + \sigma_i^2 + \sigma_j^2}} \cdot exp(-\frac{(\mu_i - \mu_j)^2}{2(2\beta^2 + \sigma_i^2 + \sigma_j^2)}) \tag{8}$$

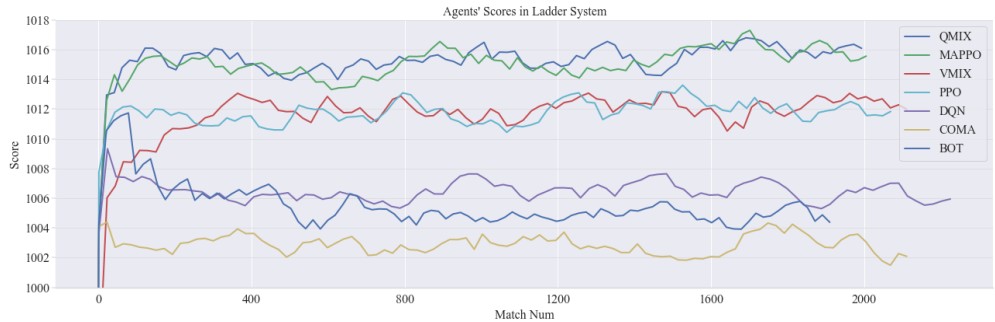

Figure 11: Comparing agents on the ladder system. The scenario is *st_t2p2*.

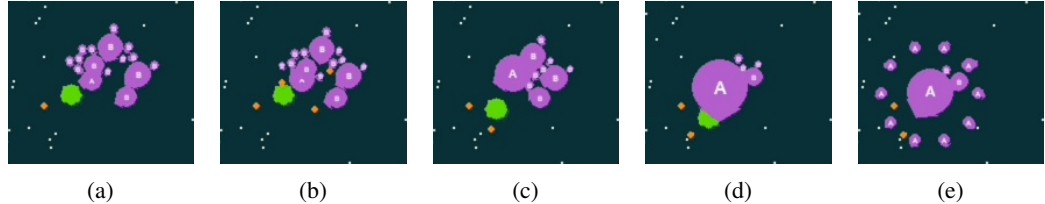

Figure 12: Screenshots of cooperating to eat the thorn ball. (a) The clone ball A and B get closed to each other. (b) The clone ball B eject to the A. (c) The clone ball A eat the spore and grow bigger. (d) The clone ball A eat several clone balls from teammates and grow big enough to eat the thorn ball. (e) The clone ball A eat the thorn ball.

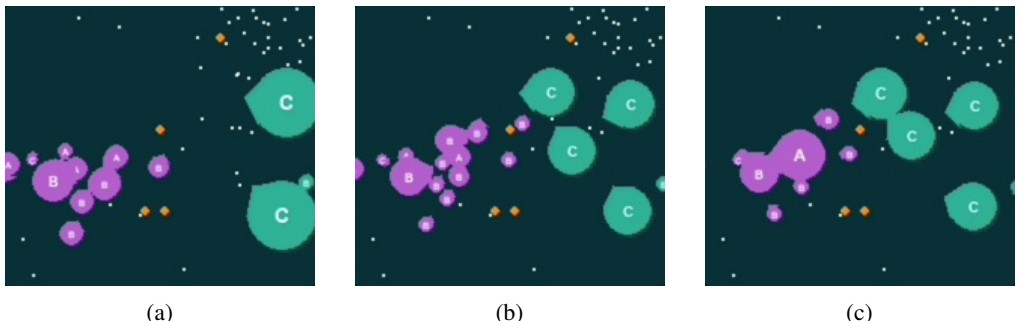

Figure 13: Screenshots of competition between different agents. (a) The green clone balls find the purple clone balls nearby. (b) The green clone balls split to eat the purple ones. (c) The purple clone balls gather together to avoid being eaten by the green clone balls.

where contestant $i$ is assumed to exhibit a performance $p_i \sim \mathbb{N}(p_i; \mu_i, \sigma_i^2)$, contestant $j$ is assumed to exhibit a performance $p_j \sim \mathbb{N}(p_j; \mu_j, \sigma_j^2)$, and $\beta^2 = \left(\frac{\sigma}{2}\right)^2$

Figure 11 shows the ladder system including all the trained agents in Section 5. Most of the contestants have played over 1500 games with other contestants. The ladder system ensures that each contestant can match opponents with similar game levels as much as possible while retaining the possibility of matching opponents with higher or lower game levels.

## D  PERFORMANCE

In this section, we post some screenshots to show the actual performance of the agent in the game, including the cooperation with teammates and the competition with opponents.

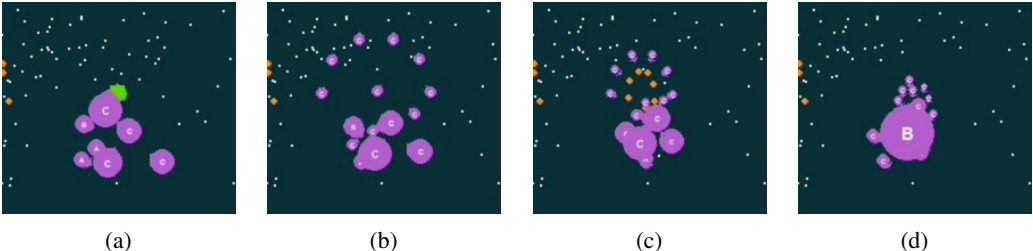

Figure 14: Screenshots of cooperation between different agents. (a) The clone ball C is ready to eat the thorn ball. (b) The clone balls split into many pieces. (c) The clone balls eject to the center of the agents. (d) The clone balls finally merge into a big one.

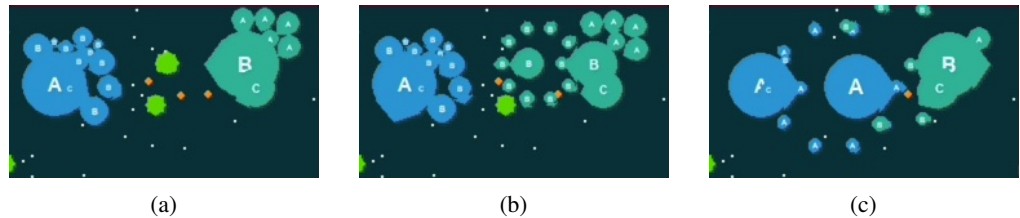

Figure 15: Screenshots of competition between different agents. (a) The bule and green agents are competing for the thorn ball. (b) The green clone balls split to eat the thorn ball. (c) The blue clone balls split and eat the green clone balls and win more score.

