# OpenReview forum: "GoBigger: A Scalable Platform for Cooperative-Competitive Multi-Agent Interactive Simulation"
_ICLR.cc/2023/Conference — ICLR 2023 poster_

### Official Review · Reviewer_DcyR · 2022-10-22

**Confidence:** 4
**Correctness:** 3
**Technical Novelty And Significance:** 2
**Empirical Novelty And Significance:** 2
**Recommendation:** 5

**Clarity, Quality, Novelty And Reproducibility:**

The contribution of this work is limited in terms of novelty and originality.

The authors provide limited information in terms of design and implementation of the algorithms used in the implementation. For this reason, the reproducibility of this work is somehow limited.

There are limited details about the implementation of the platforms. The information related to the selection of the hyperparameters of the algorithms is also somehow limited.

Minor issues include the readability of the plots (see for example Figure 3), which should be improved. At the moment, it is very hard to judge the performance results presented in the figures.


**Strength And Weaknesses:**

Strengths
+ The authors have developed a platform that might be beneficial for the community at large.

Weaknesses
- It is difficult to identify a clear novel technical contribution in this work. For sure, the reviewer commends the fact that the authors made a new environment available to the research community.
- The authors do not clarify and discuss the types of studies that are enabled by this platform. Which questions will the community be able to answer through it? What are the limitations of the platform? What are the specific novel insights that a researcher can derive from adopting this testbed/environment?
- The goal of the evaluation presented in this work should be clarified. In fact, this appears as a paper presenting a platform, but most of the paper is devoted to the analysis of existing algorithms used for playing the game supported by the environment. The authors should perhaps rethink the scope and the focus of the paper. At the moment, unfortunately it does not appear sufficiently focussed.
- The authors say that the main contribution is the study of collaborations inside a team and across teams (and competition). This is not clearly discussed in the paper in my opinion. I would also argue that other environments allow for this type for analysis - for example the Hide and Seek game is performed by teams and not by single agents (it is an environment where the agents collaborate and compete at the same time).


**Summary Of The Paper:**

This paper presents GoBigger, a platform for studying cooperative and competitive behavior at large scale. The game is inspired by Agar. Even if the design of an environment like that described in the paper might be valuable for the community, it is difficult to identify clear technical contributions in this work. The authors tested the platform using implementations of algorithms, namely DQN and PPO for single-agent algorithms and QMIX, MAPPO, COMA and WMIX for multi-agent environments. The authors showed the results of a comparative analysis of the algorithms but the results appear rather standard and not particularly insightful. It is also worth noting that the implementation of the system itself appears to be rather standard in terms of design and implementation choices.


**Summary Of The Review:**

Overall, this work makes a contribution to the research community in terms of the platform itself, but its novelty is somehow limited both in terms of the design and implementation of the environment itself and the evaluation of the algorithms presented in the paper. In general, it is difficult to understand the specific insights that it is possible to obtain from adopting it compared to other existing environments.

---

> ### Author Response · Authors · 2022-11-15
> **Thanks for your insightful comments and we would like to address your concerns.**
>
> > Q1: The authors do not clarify and discuss the types of studies that are enabled by this platform. Which questions will the community be able to answer through it? What are the limitations of the platform? What are the specific novel insights that a researcher can derive from adopting this testbed/environment?
>
> A1: In the existing reinforcement learning environment, most of the environments are constructed based on the competition between two teams. They cannot contribute to more complex game environments, such as more teams, stronger cooperation and confrontation. However, in more complex multiplayer games, it is difficult to find a Nash equilibrium or there is no Nash equilibrium at all. For example, competition among multiple enterprises in a certain field, and games among multiple countries, their relationship may constantly change in cooperation and competition in order to seek longer-term interests. With GoBigger, researchers can learn how to find a good strategy in non-zero-sum games.
> * GoBigger has no restrictions on operating systems, and installation is simple and lightweight. Both the environments and baselines are open-sourced.
> * GoBigger provides multiple sub-environments, which can cover game situations with different numbers of teams and players. It supports research on cooperative competition scenarios, swarm intelligence, strategic diversity, safe marl, etc. At the same time, we welcome researchers to explore more interesting research content on GoBigger.
>
> > Q2: The goal of the evaluation presented in this work should be clarified. In fact, this appears as a paper presenting a platform, but most of the paper is devoted to the analysis of existing algorithms used for playing the game supported by the environment. The authors should perhaps rethink the scope and the focus of the paper. At the moment, unfortunately it does not appear sufficiently focussed.
>
> A2: GoBigger is a brand-new reinforcement learning environment, so we spent a considerable amount of time introducing the environment itself, and provided an algorithm implementation to prove the correctness of the environment itself. We introduced the related content of reward in Section4.3, and proposed 3 methods to improve reward to help training. Then we pointed out that the reward method cannot measure the generalization of the agent, so we proposed in Section 6.3 to evaluate the performance and generalization of the agent through the ladder system. The ladder system will be released soon and we will regard it as one of the most important evaluation for agents.
>
> > Q3: The authors say that the main contribution is the study of collaborations inside a team and across teams (and competition). This is not clearly discussed in the paper in my opinion. I would also argue that other environments allow for this type for analysis - for example the Hide and Seek game is performed by teams and not by single agents (it is an environment where the agents collaborate and compete at the same time).
>
> A3: Indeed the Hide and Seek is an excellent setting. But we think the environment itself is still too simple, both in terms of the state of the environment and the number of actions. GoBigger provides a more complex environment, supports a larger number of agents and teams, and the action space of each agent is large enough. This can be seen in Section 4. And the behavior of cooperation and competition is more complicated in GoBigger, because the agent needs to perform long sequences of actions to complete the cooperation within the team well. For example, when completing a split cooperation, the agent needs to eject spores towards its teammates first, and then wait for a short period of time before splitting towards the teammates, so as to gather its own size on the teammates.

---

> > ### Comment · Reviewer_DcyR · 2022-12-04
> > **Thanks for the reply**
> >
> > Thanks for all the clarifications. However, I have still concerns about the actual contribution of this work (and its potential impact). The evaluation itself is not convincing in my opinion.

---

### Official Review · Reviewer_tWZb · 2022-10-24

**Confidence:** 5
**Correctness:** 3
**Technical Novelty And Significance:** 1
**Empirical Novelty And Significance:** 2
**Recommendation:** 3

**Clarity, Quality, Novelty And Reproducibility:**

- Novelty is a concern for this paper. It is not clear why extending from a two-team simulator to multi-team simulator is a key challenge. How this simulator can impact real-world MARL applications is also not clear.

+ The presentation is clear.

**Strength And Weaknesses:**

Strength:

+ Multi-team multi-agent simulators are important to evaluate RL in cooperative-competitive environments.
+ The introduced simulator offers various features to customize and interact with the simulations.

Weakness:

- The technical advances or practical novelties are not clear. It seems the main contribution is the design of multi-team simulation. However, why extending a two-team multi-agent simulation to a multi-team simulation is challenging?

- The scenario of the simulation is relatively simple. It seems the simulation only considers the agent moving directions but does not consider the agent velocity or the dynamics of the agent.

- It is not clear how this simulation can benefit real-world multi-agent applications. Other than the multi-player Agar game, what do real-world applications share characteristics similar to this simulation?

- While experiments generally show that this simulator can be used to evaluate multiple RL and MARL methods, how do the experiments demonstrate this simulator is better than the previous MARL simulators?



**Summary Of The Paper:**

This paper discusses a new multi-agent simulator that is designed to evaluate cooperative-competitive multi-agent reinforcement learning. The contributions seem to focus on simulating multiple teams with various features (e.g., choices of using local individual observations or team observations).

**Summary Of The Review:**

While designing a multi-team multi-agent simulator is important to benchmark MARL methods, novelty of the simulator and its impacts on real-world MARL applications are not well justified.

---

> ### Author Response · Authors · 2022-11-15
> **Thanks for your insightful comments and we would like to address your questions.**
>
> > Q1: The technical advances or practical novelties are not clear. It seems the main contribution is the design of multi-team simulation. However, why extending a two-team multi-agent simulation to a multi-team simulation is challenging?
>
> A1: In the process of two-team zero-sum game, there is a Nash equilibrium that makes the player's strategy optimal. However, in more complex multiplayer games, it is difficult to find a Nash equilibrium or there is no Nash equilibrium at all, and we call that Non-zero-sum game. Non-zero-sum games are common in economics, political science, and modern warfare. For example, put money into a company. This doesn't become a clean situation where they will either gain or lose, because it's possible for the company to use their investment to gain overall value. It's possible for everyone to get wealthier than when they started. And the competition between multiple companies in a certain field, as well as games between multiple countries. The relationship between different enterprises and even different countries may constantly change in cooperation and competition in order to seek longer-term interests.
>
> > Q2: The scenario of the simulation is relatively simple. It seems the simulation only considers the agent moving directions but does not consider the agent velocity or the dynamics of the agent.
>
> A2: The characteristic of Agar games is that the scene is simple, because it pays more attention to the interaction between players. In GoBigger, we simulate the principles of real physics in motion, such as collision mechanism, speed and velocity decay and acceleration. As can be seen from the action space, the agent changes the direction and acceleration of the movement by setting (x, y), and then the speed changes accordingly. Meanwhile, GoBigger provides the two skills of splitting and ejecting spores to strengthen the cooperation and difficulty of competition between different agents, which can be found in Section 3.1.
>
> > Q3: It is not clear how this simulation can benefit real-world multi-agent applications. Other than the multi-player Agar game, what do real-world applications share characteristics similar to this simulation?
>
> A3: GoBigger simulates multi-team (multi-player) games very well. For example, at the micro level, we can regard each agent as a single-celled organism in the environment, which grows and develops by ingesting nutrients (spore) in the environment. In addition, it can split to produce new cells and eject spores to aid its movement in the environment. The ultimate purpose of the cell is to occupy all the nutrient resources in the environment and exclude other organisms in the environment. This is consistent with the design concept and execution logic of GoBigger.
>
> > Q4: While experiments generally show that this simulator can be used to evaluate multiple RL and MARL methods, how do the experiments demonstrate this simulator is better than the previous MARL simulators?
>
> A4:   GoBigger emphasizes cooperation and competition between multiple teams. Prior to this, only Neural MMO supported this type of environment.
> * Neural MMO is a virtual simulator and it's not verified in its game mechanics. However, GoBigger is an RL environment abstracted from the real game environment.  And numbers of great videos and the experience of professional players proves that there must be an optimal strategy for this game mechanism, and we can discover diverse multi-agent strategic behavior.
> * We found that in Neural MMO intra-team cooperation is negligible and inter-team competition is not nesserary, because the goal of agents is to complete the resource gathering or skill learning tasks. While GoBigger puts more emphasis on the cooperation within the team and the competition between the teams. And it is proved through experiments that in the case of losing the cooperation in the team, the performance of the agent will drop sharply. According to the baseline provided by Neural MMO, it is difficult to see the cooperation and confrontation between agents in the game, and the algorithm is more inclined to complete the survival and development that can be achieved by a single agent. Therefore, we think GoBigger fills this gap, that is, it pays more attention to the cooperation and confrontation process between multi-team and multi-player.
> * Neural MMO only provides a simple baseline, and there is no ladder system, it is difficult to verify the generalization of the agent in the face of different opponents, and there are no relevant indicators to measure the strength of cooperation. And GoBigger provides a new ladder system to help players verify their agents' performance. In addition, GoBigger also provides a series of indicators (such as the number of clone balls eaten by agents, the number of spore balls ejected by agents) to measure the strength of cooperation and confrontation in the game.

---

### Official Review · Reviewer_3xPp · 2022-10-25

**Confidence:** 4
**Correctness:** 3
**Technical Novelty And Significance:** 3
**Empirical Novelty And Significance:** 3
**Recommendation:** 6

**Clarity, Quality, Novelty And Reproducibility:**

Clarity: The paper is written sufficiently clear. The game dynamics are well-introduced and the configuration options are understandable. The appendix also includes more practical examples of using the environment.
Quality: The proposed environment is a sensible multi-agent and multi-team game. It builds on an established game and enables to study coordination and competition.
Novelty: At this point, the novelty compared to Neural MMO is not completely clear given the points I made in the other sections. It should be better argued in the paper why and how coordination is better.
Reproducibility:  The empirical results are documented sufficiently well and since the authors open-source their code, the results can be reproduced.

**Strength And Weaknesses:**

Strenghts:
* A sensible extensions towards multiple teams of an established environment
* Empirical evaluation for the new environment alone is quite exhaustive, showing that the environment fosters coordination among teams as winning strategies

Weaknesses:
* The paper does not motivate sufficiently well how/why coordination is not evolving in the main competing environment Neural MMO. The claim should be better explained / backed up with further analyses. If there is a theoretical point to be made, it would fit nicely in the motivation and related work, but pointers to empirical results (showing sufficiently more coordination for the new environment) would be helpful as well.

**Summary Of The Paper:**

The paper proposes a novel multi-agent RL environment which extends existing game Agar with multiple teams, thereby enabling M x N style games. A single agent in the game manages up to 16 growing cloned balls, which need to be increased in size to gain more rewards. The resulting action space has a discrete- (type of action - move, eject, split) and continuous component (direction of movement). The approach is evaluated for the different configurations of the environment for both single agent as well as multi-agent RL algorithms. The results show the tendency for cooperation. Lastly, the authors also conduct an ablation study.

**Summary Of The Review:**

The proposed environment is interesting. In essence, the game enables the study of cooperation and competion between multiple teams, and has a well-defined logic and resulting game complexity. The comparison to other environments is quite clear. The paper provides a well-structured oview of the main competitors and builds its case on enabling more than two teams to compete and enabling the better study of cooperation. To this end, it is difficult for me to see if the claim about lack of cooperation for main competitor Neural MMO is evident. Is it that Neural MMO is to easy to drive coordination? Or maybe the other extreme, is the logic so difficult to allow for complex behaviour with current algorithms?  The conducted evaluation for the proposed environment shows that teams are coordinating to win and not merely try to fight individual opponents by themselves, which shows that coordination is essential for winning. Is this a strategy evolving over time or occuring rather fast? To sum up, I see that the environment has its merits, but am not (yet) conviced of the novelty.

---

> ### Author Response · Authors · 2022-11-15
> **Thanks for your insightful comments and we would like to address your questions.**
>
> > Q1: The paper does not motivate sufficiently well how/why coordination is not evolving in the main competing environment Neural MMO. The claim should be better explained / backed up with further analyses. If there is a theoretical point to be made, it would fit nicely in the motivation and related work, but pointers to empirical results (showing sufficiently more coordination for the new environment) would be helpful as well.
>
> A1: Neural MMO is the first environment in the field of reinforcement learning to support multi-team and multi-player games. Compared with Neural MMO, GoBigger emphasizes cooperation and competition between different teams. However, we found that in Neural MMO,
> * intra-team cooperation is negligible, and inter-team competition is not important, as the goal of Neural MMO is to complete the tasks of resource collection or skill learning. GoBigger is an RL environment abstracted from the real game environment. Numbers of videos and the experience of professional players proves that cooperation in this game is necessary. It is easy to figure out how to cooperate with teammates and how to competite against other taems. Meanwhile, Ours experiments which in Figure 5 also demonstrate that in GoBigger, cooperation will yield higher benefits than non-cooperation. To further demonstrate that Gobigger is an environment that emphasizes cooperation, we supplement the comparative experiments between team rewards and individual rewards. When agents used team rewards, they performed better than individual rewards.
> * In addition, according to the baseline provided by Neural MMO, it is difficult to find cooperation and competition between agents in the game, and the algorithm is more inclined to complete the survival and development that can be achieved by a single agent. GoBigger provide more algorithms and baselines in environments such as `st_t2p2`, `st_t3p2` and `st_t4p3`, and the replays could show the cooperation and competition in the game.
>
> Therefore, we are sure that GoBigger will be helpful for researchers to focus more on cooperation and competition in multi-agent environments.

---

### Decision · Program_Chairs · 2023-01-20

**Decision:**

Accept: poster

**Justification For Why Not Higher Score:**

This is a good benchmark that is worth publishing. However, the impact of the benchmark will be decided by the community in the near future so I would leave it as a poster.

**Justification For Why Not Lower Score:**

There is no single strong reason to reject the paper. The only negative complaint is about the novelty but there are not enough MARL benchmarks out there so I don't mind this benchmark getting published. It is interesting enough for the community to use it in their research.

**Metareview: Summary, Strengths And Weaknesses:**

This paper proposes a new multi-agent RL environment which has the possibility of having many teams each needing to cooperate within the team and compete with other teams. Authors argue that the cooperation and competition aspects are better highlighted in this proposed benchmark than in the existing benchmarks.

One of the main complaints of the reviewers is about the novelty. However, given the lack of good benchmarks for MARL, I do not see novelty as a big concern. Given that there were no complaints about the correctness of the paper, I recommend an acceptance.





**Note From Pc:**

if the above contains the word "oral" or "spotlight" please see: "oral" presentation means -> notable-top-5% and "spotlight" means -> notable-top-25%. As stated in our emails, we are disassociating presentation type from AC recommendations